# Gastrointestinal Dysfunction in Parkinson’s Disease: Current and Potential Therapeutics

**DOI:** 10.3390/jpm12020144

**Published:** 2022-01-21

**Authors:** Myat Noe Han, David I. Finkelstein, Rachel M. McQuade, Shanti Diwakarla

**Affiliations:** 1Gut-Axis Injury and Repair Laboratory, Department of Medicine Western Health, University of Melbourne, Melbourne, VIC 3021, Australia; cherish.han@unimelb.edu.au (M.N.H.); lakshmi.diwakarla@unimelb.edu.au (S.D.); 2The Florey Institute of Neuroscience and Mental Health, Parkville, VIC 3010, Australia; 3Australian Institute of Musculoskeletal Science (AIMSS), Melbourne, VIC 3021, Australia; 4Parkinson’s Disease Laboratory, The Florey Institute of Neuroscience and Mental Health, Parkville, VIC 3010, Australia; david.finkelstein@florey.edu.au

**Keywords:** Parkinson’s disease, gastrointestinal dysfunction, enteric nervous system, enteric neuropathy, constipation, gastroparesis, dysphagia, unmet therapeutic need

## Abstract

Abnormalities in the gastrointestinal (GI) tract of Parkinson’s disease (PD) sufferers were first reported over 200 years ago; however, the extent and role of GI dysfunction in PD disease progression is still unknown. GI dysfunctions, including dysphagia, gastroparesis, and constipation, are amongst the most prevalent non-motor symptoms in PD. These symptoms not only impact patient quality of life, but also complicate disease management. Conventional treatment pathways for GI dysfunctions (i.e., constipation), such as increasing fibre and fluid intake, and the use of over-the-counter laxatives, are generally ineffective in PD patients, and approved compounds such as guanylate cyclase C agonists and selective 5-hyroxytryptamine 4 receptor agonists have demonstrated limited efficacy. Thus, identification of potential targets for novel therapies to alleviate PD-induced GI dysfunctions are essential to improve clinical outcomes and quality of life in people with PD. Unlike the central nervous system (CNS), where PD pathology and the mechanisms involved in CNS damage are relatively well characterised, the effect of PD at the cellular and tissue level in the enteric nervous system (ENS) remains unclear, making it difficult to alleviate or reverse GI symptoms. However, the resurgence of interest in understanding how the GI tract is involved in various disease states, such as PD, has resulted in the identification of novel therapeutic avenues. This review focuses on common PD-related GI symptoms, and summarizes the current treatments available and their limitations. We propose that by targeting the intestinal barrier, ENS, and/or the gut microbiome, may prove successful in alleviating PD-related GI symptoms, and discuss emerging therapies and potential drugs that could be repurposed to target these areas.

## 1. Introduction

Parkinson’s disease (PD) is the second most frequent neurological disease after Alzheimer’s disease, and affects 1% of the population above 60 years of age worldwide [1]. Clinical hallmarks of the disease include resting tremor, rigidity, bradykinesia, and postural instability, which is primarily attributed to the progressive degeneration of dopamine-containing neurons in the substantia nigra pars compacta (SNpc) [2,3,4]. In the majority of PD cases, the presence of Lewy bodies (LB) in the SNpc, which contain aggregates of the protein alpha-synuclein, serve as a pathological hallmark of the disease. However, LBs are also found in other regions of the central nervous system (CNS), such as the peripheral and enteric nervous system (ENS) [5,6], and may play a role in the onset of non-motor symptoms associated with the disease. 

Most PD patients develop some form of autonomic disorder involving the cardiovascular, genitourinary, thermoregulatory, and gastrointestinal (GI) systems [7]. GI dysfunctions are a common occurrence in PD patients and can include symptoms such as dysphagia (difficulty swallowing), gastroparesis (slowed gastric emptying), and chronic constipation. Due to limitations associated with methods of detection, studies being performed in heterogenous patient populations, and a lack of patient reporting, the exact prevalence of GI dysfunctions in PD patients is difficult to determine. However, it is clear that some forms of GI dysfunction precede the onset of motor deficits by decades [8] and greatly impact on patient quality of life.

Gut dysfunction contributes directly to the morbidity of PD, and complicates clinical management. Weight loss has been found to precede PD diagnosis by years, and is associated with disease severity as well as duration [9], whilst low body mass index has a significant impact on the life prognosis of PD patients, particularly males [10]. Nausea, resulting from delayed stomach emptying, is thought to contribute to weight loss, whilst delayed colonic transit time due to poor motility causes harder stools and constipation by increasing the absorption of water. 

Whilst motor symptoms can be managed using levodopa—a dopamine prodrug—GI symptoms often remain unaffected, suggesting that the loss of dopamine does not drive GI symptom onset [11,12]. The presence of LBs in the ENS has led to the suggestion that enteric neuronal loss and/or dysfunction may contribute to the development of GI symptoms. The ENS is a subdivision of the autonomic nervous system, and together with its CNS connections it forms the primary control centre for GI function [13]. Beginning in the esophagus and extending down to the anus, the ENS is embedded in the lining of the GI tract and interacts with the CNS through the vagus and pelvic nerves, as well as sympathetic (prevertebral ganglia) connections that regulate contraction, relaxation, secretion, and absorption throughout the GI tract [13].

Gut biopsies from PD patients show evidence of enteric neuronal loss and alteration in the expression of important neurotransmitters, such as vasoactive intestinal peptide (VIP) and dopamine [5,14,15,16,17,18]. Moreover, altered gut barrier function and reduced expression of tight junction (TJ) proteins that mediate paracellular transport (the passage of molecules between adjacent epithelial cells in the gut) have been implicated in PD-related gut dysfunction [19,20]. In addition, the gut microbiome has gained much attention in PD with accumulating evidence linking microbial dysbiosis to PD symptomology and pathophysiology [21]. Thus, PD-induced damage at the level the ENS may be a key contributor to the development of GI symptoms. 

The conventional management of PD-induced GI complications, such as dietary interventions and exercise generally do not provide long-term efficacy for PD patients, and approved compounds, such as dopamine antagonists and selective 5-hydroxytryptamine/serotonin 4 (5-HT_4_) receptor agonists, have demonstrated limited efficacy. Thus, identification of potential targets for novel therapies to alleviate PD-induced GI dysfunctions are essential to improve clinical outcomes and quality of life amongst PD sufferers. This review describes the most common GI symptoms that affect individuals suffering from PD, and the impact they have on quality of life and disease management. Given the limitations of current treatments, we propose a novel approach to the alleviation of GI symptoms by highlighting the intestinal barrier, ENS, and gut microbiome as potential therapeutic avenues, and expand on current emerging therapies and propose several therapeutics that may prove effective through drug repurposing.

## 2. Gastrointestinal Dysfunction in Parkinson’s Disease 

The clinical manifestations of GI dysfunctions in PD include oral and dental disorders, sialorrhea (drooling), dysphagia, gastroparesis, malabsorption, constipation, and defecatory dysfunction [22,23,24,25,26,27,28]. Unlike motor symptoms, which only become apparent on formal diagnosis in the established disease (when degeneration of 50–70% of dopaminergic neurons in the CNS has occurred) [29,30,31,32], GI symptoms have been reported to arise up to 10 years prior to clinical diagnosis. Although the time of onset and severity of GI complications can vary, almost every PD patient suffers from GI symptoms during the natural course of the disease [33,34,35,36,37]. Most PD-related GI symptoms, such as dysphagia and constipation, are a result of abnormal motility of the GI tract. These symptoms not only impact on patient quality of life, but some, particularly dysphagia and gastroparesis, can worsen motor symptoms through the inadequate absorption of oral anti-PD medications. Despite vigorous attempts to understand the pathophysiological underpinnings of these manifestations over the past two decades, GI dysfunction in PD patients is not well understood and lacks effective treatments [38]. 

### 2.1. Dysphagia

Dysphagia is primarily caused by one of the cardinal symptoms of PD, bradykinesia, which reduces control of the oral, pharyngeal, and esophageal cavities [26,39,40]; however, several cortical regions also contribute to swallowing difficulties [41,42,43]. The proportion of PD patients experiencing dysphagia ranges from 9–77% [27,44]. This wide range in prevalence is thought to result from the combined effect of patients being unaware that they have symptoms, and the lack of established diagnostic criteria to assess the presence and severity of dysphagia symptoms. Dysphagia is responsible for sialorrhea, one of the most common GI complications reported in PD patients, with up to 75% of patients indicating mild to severe symptoms [39,45]. Although drooling is commonly associated with the excessive production of saliva, a recent study has shown that patients with PD produce less saliva than healthy controls [26]. The accumulation of saliva and the subsequent manifestation of drooling in patients with PD is caused by a combination of disease features, which include dysphagia, a stooped posture, and a tendency towards an open mouth [39]. 

Whilst there is no correlation between the severity of dysphagia and the clinical progression of PD, there is a clear link between dysphagia and an increased risk of mortality. Dysphagia can cause additional complications such as malnutrition and subsequent weight loss and dehydration, and aspiration pneumonia (estimated to account for 70% of the mortality rates among PD patients) [46,47,48,49]. 

### 2.2. Gastroparesis

Gastroparesis is characterized by feeling full after only a small amount of food (early satiety), and the sensation of having food in the stomach (postprandial fullness), which can lead to weight loss, malnutrition, abdominal pain, nausea, vomiting, and bloating [50,51]. Based on scintigraphy studies, approximately 70–90% of PD patients suffer from gastroparesis [50,52]; however, this high prevalence most likely includes asymptomatic manifestation, as patient reporting indicates that observed features of gastroparesis are present in 25–45% of patients [53]. Gastroparesis can be present in both early and late stages of the disease, and unlike dysphagia, the severity of gastroparesis correlates with the severity of motor impairment [52]. 

The pathophysiological abnormalities of gastroparesis in PD include central and enteric components, with the dorsal motor nucleus of the vagus involved early in the course of disease [54]. The vagus nerve interacts with the ENS to cause neuromuscular dysfunction, which can result in antral hypomotility and/or dysfunction of the pyloric sphincter [55]. These changes in normal function result in delayed stomach emptying that can significantly impact on the control of motor symptoms by delaying the absorption of drugs, such as levodopa, in the small intestines [22,56]. Compounding this dilemma, levodopa treatment can lead to gastroparesis [50], making it difficult to directly attribute gastroparesis to PD. Nevertheless, it is obvious that treatment for this GI symptom is needed to both improve quality of life and control PD symptoms overall.

### 2.3. Constipation

Constipation, caused by delayed colon transit or the irregular contraction of the voluntary sphincters during defecation [57], is another common GI symptom in PD patients [27,58]. The estimated prevalence of constipation in PD ranges from 7% to 70%, depending on the specific definition [59,60]. According to the most frequently used definition for constipation (<three bowel movements per week) [61], the median prevalence is estimated to be 50% [62]. Similar to other PD-related GI symptoms, the prevalence of constipation within the PD community is difficult to determine due to the varying diagnostic criteria and methods used to assess the symptom. Moreover, drugs used to control motor symptoms, particularly anticholinergics, can both cause and exacerbate constipation [63].

In addition to the vagal connection between the CNS and ENS, the presence of synucleinopathy in the submucosal plexus and myenteric plexus of the colon may also directly affect the action of intestinal muscles that function to move material through the intestines [64]. Constipation is thought to occur early in the course of disease, and it has been reported that it could serve as a predictor, with individuals suffering from long-term constipation having a 2–5 times higher risk of suffering from PD later in life [65,66]. 

## 3. Current Treatments for Gastrointestinal Dysfunction in Parkinson’s Disease 

Given the impact GI complications have on patient quality of life and drug efficacy, it is important to control symptom severity to enable clinical management of the disease overall. Currently, many treatments involve dietary interventions, physical therapies, and in some cases, drug interventions that target muscle relaxation and contraction of the GI tract (Figure 1). These treatment options have been somewhat successful in alleviating the symptoms, but many have unwanted side effects or offer only short-term relief.

### 3.1. Dysphagia

Initial treatments for dysphagia primarily revolve around managing food intake, such as requesting patients to chew food more thoroughly, eating smaller portions, and trying foods with softer textures to make it easier to swallow. However, these interventions may not be suitable for all patients, particularly those with more severe symptoms. Swallowing therapy as a rehabilitative treatment has shown promising results [24,67], with a randomised and blinded control study demonstrating that expiratory muscle strength training improved swallowing safety and hyolaryngeal function during swallowing [68]. In addition, video-assisted swallowing therapy, which allows patients to visualize their swallowing patterns, has proven to be effective [69]. This positive outcome is mainly due to PD patients responding better to visual information, and not having to rely on proprioceptive signals and remembered visual information [69,70]. 

Surgical treatments such as cricopharyngeal sphincterotomy have been reported to yield excellent results [71,72]. However, most studies have involved small samples sizes, and surgical intervention may not be appropriate for all forms of dysphagia (oropharyngeal dysphagia vs. oesophageal dysphagia). Neuromuscular electrical stimulation (NMES) has also been trialled and compared to traditional therapies for dysphagia (logopedic treatment by a speech therapist) in a randomized control trial where NMES was used to facilitate contraction of the suprahyoid muscles. Interestingly, both the traditional method and NMES significantly improved patient quality of life scores; however, there was no significant difference between the two [73]. 

The outcomes of pharmacological interventions with respect to dysphagia vary substantially between studies and patients. In general, dopaminergic drugs are helpful for dysphagia in PD [47,74,75]; however, some studies showed no improvement [76,77], while others showed equivocal results [78,79,80]. Dysphagia in patients with early stages of PD can be ameliorated by implementing dopaminergic treatment during the first few years after disease onset [22]. However, dysphagia is usually underdiagnosed in these patients due to poor self-awareness of the symptoms [81,82]. Therefore, early detection and individualised intervention are crucial for improving quality of life in patients. 

### 3.2. Gastroparesis

In general, the therapeutic options for gastroparesis are limited and there is currently no gold standard for the treatment of gastroparesis in PD. Management of gastroparesis in PD predominantly involves changes in diet, antiemetics, prokinetics, and in severe cases, feeding tubes or parenteral nutrition [83,84]. Pharmacological treatment of gastroparesis is limited; however, several clinical trials focusing specifically on the management of PD-related gastroparesis have recently been conducted using 5-HT_4_ receptor agonists, erythromycin, and motolin receptor agonists, which indicate the importance of managing this symptom.

Dietary interventions can help reduce symptoms and improve nutritional deficiencies, and are used as the first line of management for gastroparesis with different aetiologies [85]. These interventions include eating small frequent meals consisting of a low-fibre and low-fat diet of, predominantly, liquids and protein supplements. In such cases, the identification and correction of nutritional deficiencies needs to be considered, such as zinc, iron, selenium, folate, B12, vitamin C and vitamins A, D, K and E. For more severe gastroparesis, a liquid diet, a jejunal feeding tube, gastric electrical stimulation, total parenteral nutrition, or other approaches, may be necessary [85].

With regards to pharmaceutical intervention, to date, only domperidone has proven to be both effective and safe in the treatment of gastroparesis [86]. Domperidone is a peripheral dopamine receptor 2 antagonist, that functions by blocking the inhibitory effect of dopamine, resulting in the stimulation of muscle contraction. In patients with gastroparesis, domperidone increases oesophageal peristalsis and facilitates gastric emptying by augmenting gastric peristalsis and improving antroduodenal co-ordination [87,88]. 

As well as its prokinetic properties, domperidone has the added benefit of being a potent antiemetic and may increase the bioavailability of levodopa [89,90]. Furthermore, domperidone does not readily cross the blood–brain barrier, and thus the risk of developing extra-pyramidal side-effects is minimal. However, domperidone may prolong the time taken for ventricular depolarisation and repolarisation (QT interval) in predisposed patients, and has been associated with arrhythmia, cardiac arrest, and sudden death at high intravenous doses. Recent case-control studies have also indicated concerns regarding cardiotoxicity following oral administration, particularly in patients over 60 years of age [91,92].

Whilst domperidone remains the most studied prokinetic in patients with PD, other dopaminergic blockers, such as metoclopramide, have been trialled clinically. Metoclopramide is the only Food and Drug Administration (FDA)-approved medication for gastroparesis [93] and is commonly used to prevent levodopa-induced nausea and vomiting. However, metaclopromide can readily cross the blood–brain barrier and has central dopaminergic activity [94,95]. As a result, metoclopramide can precipitate unwanted side effects of excess dopamine receptor stimulation, and its use is not recommended. 

Several therapeutic interventions have shown promise for the treatment of gastropareisis, such as the selective 5-HT_4_ receptor agonists cisapride, tegaserod, and mosapride. Although these drugs were found to alleviate PD-induced gastroparesis, tegaserod and cisapride were found to have toxic cardiovascular effects [96,97], while mosapride can induce abdominal pain, nausea, diarrhea and sometimes constipation.

### 3.3. Constipation

Prolonged colon content transit time is the physiological basis of symptomatic and asymptomatic constipation [98]. Dysmotility may be related to poor fibre and fluid intake or reduced physical activity [66]. Management of constipation commonly involves interventions that are used to treat constipation in the general population, such as dietary changes, an increase in physical activity, and the use of laxatives and stool softeners. 

Lifestyle modifications, especially dietary interventions, are predominantly used as a first-line treatment for constipation. Interventions include increasing dietary fibre through the use of fibre supplements, which have been found to increase stool frequency and output compared with placebo in the general population [98]. Fibre plays an important role in the management of constipation and adequate consumption, at least 30 to 35 g daily, together with appropriate fluid intake (at least 1500 mL daily), is often encouraged [65,99]. Given that stool water content is a crucial factor for stool softening, and consequently, for stool consistency, water-soluble fibres such as psyllium or β-glucan are also recommended. These categories of fibre absorb water to become a gelatinous, viscous substance, which are fermented by bacteria in the digestive tract, promoting peristalsis and increasing stool bulk, softness and intraluminal pressure [28]. Although there is conflicting evidence for the effectiveness of soluble fibres and insoluble fibres for constipation, psyllium intake has shown promise in PD patients where 8 weeks of supplementation increased stool frequency and weight [98]. In addition, psyllium was found to improve the absorption of levodopa, thus providing more stable and higher concentrations of circulating drug [100]. However, despite advice from the World Gastroenterology Organisation to include more fibre in the diet, there is contradictory evidence regarding the use of fibre as a dietary treatment for different types of constipation. 

Physical therapies, such as abdominal massage, are known to relieve constipation by stimulating peristalsis, decreasing colonic transit time, and increasing the frequency of bowel movements [101,102,103]. Given that physical interventions are free of harmful side effects, they are often recommended as an adjunct intervention to PD patients [104]. However, qualitative research by McClurg et al. (2016) testing recruitment, retention, and the intervention’s appropriateness failed to demonstrate significant clinical improvement in stool frequency compared to lifestyle advice. Moreover, the Movement Disorder Society concluded that there was “insufficient evidence” for the beneficial effects of abdominal massage on relieving constipation and that the intervention was “investigational” [105]. Although the use of physical therapies such as abdominal massage in PD-associated constipation is contentious, the intervention is generally well tolerated, and side effects are scarce. Thus, there is room for further research.

Polyethylene glycol (PEG) and lubiprostone are first-line pharmacological interventions recommended by evidence-based medicine guidelines for the treatment of constipation due to slow colonic transit in PD [106]. PEG, also known as macrogol, is an osmotic laxative that passes through the GI tract without being absorbed and relieves constipation by drawing water in the bowel. As a result of excess water in the lumen, water content and volume of the stools in the bowel are increased, making them softer and easier to pass. Bowel movements, stool consistency and ease of defecation have been found to be significantly improved in PD patients administered twice daily with isosmotic macrogol electrolyte solution (PEG 7.3 g sachets dissolved in 250 mL of water) both in a double-blind randomized control trial [107] and in an open study [108]. Moreover, a Cochrane review found a statistically significant improvement in the number of bowel motions or successful bowel care routines per week following the use of PEG [109]. Unfortunately, abdominal bloating, vomiting, and cramps have been reported as the most common adverse effects of PEG in other investigations [110,111,112].

Lubiprostone is an intestinal chloride secretagogue that functions by selectively activating type 2 chloride channels in the apical membrane of the GI epithelium [113]. It was approved by the FDA in 2006 for the treatment of chronic idiopathic constipation in adults, and in 2008 for irritable bowel syndrome (IBS) with constipation in adult women [106]. A single placebo-controlled trial of lubiprostone for constipation associated with PD found a marked or very marked clinical global improvement in 16 of 25 (64.0%) patients receiving lubiprostrone vs. 5 of 27 (18.5%) patients receiving placebo (*p* < 0.001) alongside significant improvements in constipation rating and stools per day. Adverse events were mild, with the most common being intermittent loose stools [114]. Lubiprostone has also been found to have a prostaglandin-like action in the intestine, such that it stimulates mucin release, and has been shown to protect and repair the epithelial barrier and cell function in animal models of stress and disease [115,116]. 

Patients with PD commonly suffer from GI dysfunctions such as dysphagia, gastroparesis, and constipation. Many of these symptoms appear decades before the onset of motor symptoms. Current treatments often include physical therapies and dietary interventions. Adapted from “Adult female (anterior, lateral head, open mouth”, by BioRender.com (accessed on 17 January 2022) (2021). Retrieved from https://app.biorender.com/biorender-templates (accessed on 17 January 2022).

## 4. Potential Treatments for Gastrointestinal Dysfunction in Parkinson’s Disease

To date, there are no clinically effective neuroprotective or disease-modifying treatments that can halt PD progression. The current clinical approach for both motor and non-motor symptoms, including GI dysfunctions, focus on symptomatic management and do not aim to prevent or reverse disease progression. Given that GI dysfunction affects drug pharmacodynamics, potentially worsening motor fluctuations and further disability, it is of critical importance that therapeutics/interventions that improve GI health and function are identified. Current treatments for dysphagia, gastroparesis, and constipation result in additional side-effects, and in most cases are not recommended for long-term use, which is difficult when many PD patients require daily intervention [117]. In addition, most interventions have not been validated in the specific context of PD-induced GI dysfunction. 

## 5. Targeting Enteric Neurons

Since the introduction of Braaks staging hypothesis, which states that LBs can initiate their spread from the ENS to unmyelinated ganglionic fibres of the dorsal motor nucleus of the vagus (DMV), multiple studies have shown that alpha-synuclein has prion-like properties that allow it to spread from the ENS to the CNS [118,119,120]. The idea that the GI tract may be involved in PD pathology and progression has sparked interest in whether protecting enteric neurons could prevent GI symptoms, as well as impeding the spread of the disease to the CNS [8]. 

Deficits at the level of the ENS have been reported in PD patients. Enteric neuronal loss has been found in the ascending and descending colon of patients with PD [17], and downregulation of VIP in colonic submucosal neurons has been reported in PD patients with chronic constipation [121]. Although the extent of neuronal damage in the GI tract in patients with PD varies between studies [122], alpha-synuclein inclusions have been found in dopaminergic enteric neurons [123] and are more common in PD patients than healthy controls [5,17,123,124,125]. However, the contribution of this finding to GI symptomology remains unclear. 

Enteric neuropathy and changes in enteric neuron subpopulations have been linked to gut dysfunction in other pathological conditions, including inflammatory bowel disease (IBD), chronic intestinal pseudo-obstruction, and Hirschsprung’s disease [126,127,128]. Thus, it is conceivable that targeting the ENS could be beneficial in impeding gut symptoms and potentially the pathological progression of PD in the CNS.

### 5.1. 5-Hydroxytryptamine 4 (5-HT_4_) Receptor Agonists

5-HT is predominantly found in enteroendocrine cells located on the mucosal surface of the gut [129] where it is involved in a number of physiological roles including motility, secretion, and mucosal growth and maintenance [130]. Of the various 5-HT receptors, the 5-HT_4_ receptor plays a prominent role in releasing several neurotransmitters, most importantly, acetylcholine, the major excitatory neurotransmitter in the GI tract, to modulate propulsive patterns of gut motility [129,131,132].

Prucalopride is a highly selective 5-HT_4_ receptor agonist that is approved for the treatment of chronic constipation, primarily in elderly patients [133]. Prucalopride, as well as other 5-HT_4_ agonists such as velusetrag and naronapride, have been proven to be effective in various conditions associated with constipation with impressive safety and tolerance profiles [134,135]. In contrast to studies in healthy human subjects where the prokinetic effect of prucalopride was localized to the colon [136], in patients with constipation, prucalopride also aided with gastric emptying [137,138]. 

Prucalopride has additional properties that make it a novel therapeutic candidate for the treatment of GI dysfunctions that involve neuronal injury/loss. Prucalopride has been shown to protect primary human enteric neurospheres from oxidative stress injury in culture and protect both excitatory and inhibitory neuron populations [139] that are important for the contraction and relaxation of the GI tract (Figure 2, panel A). Although prucalopride has not been extensively trialled in PD patients for constipation, the small studies that have been performed are promising [140,141]. In addition, the neuroprotective role of prucalopride widens its role, and other 5-HT_4_ receptor agonists, as a potential therapeutic for GI dysfunctions that involve neuron damage/loss, such as that seen in PD. 

### 5.2. Flavonoids

Flavanoids are group of plant metabolites that are found in a variety of fruits and vegetables. Flavanoids modulate several signalling pathways and are known to have antioxidative, anti-inflammatory, anti-mutagenic, and anti-carcinogenic properties. Depending on their structure, flavanoids can be subdivided into subgroups, which include flavanols, flavanones, flavones, flavonols, isoflavonoids, and anthocyanidins [142].

Flavanols are known for their antioxidant properties and occur abundantly in fruits and vegetables, tea, and red wine. The most well studied flavonols are kaempferol, quercetin, myricetin and fisetin. Quercetin (3,5,7,3′,4′-pentahydroxyflavone) is one of the most potent antioxidants among the flavanoids and has been extensively studied as a potential therapeutic for neurodegenerative diseases, cardiovascular disease, and cancer. Of importance for this review, quercetin has been shown to enhance gut barrier function by increasing the expression of tight junction proteins [143] and has demonstrated neuroprotectant properties [144,145,146], which can subsequently improve motor symptoms in animal models of PD [144,147].

The beneficial effects of quercetin in PD were first identified in the 6-hydroxydopamine rat model of PD, where intraperitoneal injection of quercetin two days after lesioning for 14 days was found to protect dopaminergic neurons in the CNS and increase dopamine levels in the striatum [148]. Quercetin inhibits neuronal cell death by downregulating the pro-inflammatory cytokines NF-κB and iNOS. In addition, it can reduce alpha-synuclein fibrillization in vitro [149], and mitochondrial dysfunction (Figure 2, panel B). Thus, coupled with its role in promoting gut barrier function/integrity, quercetin may have the potential to protect enteric neurons from damage, as observed in the CNS.

Although quercetin has many beneficial effects and shows promise in vitro and in in vivo models of disease, clinical trials/studies of quercetin in PD are yet to be carried out. A possible reason for this is the limited bioavailability of quercetin when orally administered, as it is extensively metabolized in the gut [150]. In addition, the long half-life of quercetin metabolites can lead to plasma accumulation [151]. However, studies have shown that when administered with piperine, an alkaloid found in black pepper, bioavailability is enhanced by inhibiting degradation enzymes [150,152].

## 6. Targeting the Intestinal Barrier 

The GI tract is the largest mucosal surface in the body, and directly interfaces with external environmental factors. The epithelial cells that make up the epithelial lining of the intestine are held together by TJ proteins that allow the passage of ions, water, and solutes through the paracellular pathway and the movement of proteins and lipids between the apical and basolateral surfaces of the plasma membrane. This selectively permeable barrier plays an important role in protecting the body from potentially harmful substances in the lumen of the gut whilst continuing to absorb nutrients.

In addition to TJ proteins, the intestinal barrier also consists of a mucous layer, which is composed primarily of the gel-forming mucin MUC2 [153,154]. MUC2 is synthesized and secreted by epithelial goblet cells to protect the body from microbial antigens and lumen contents [155]. Reduced thickness/disruption of the mucus layer results in an increase in intestinal permeability that can lead to systemic inflammatory responses, enteric neuropathy, and potential GI dysfunction. Therefore, maintaining mucin production in the gut can improve gut health.

Increased GI barrier permeability, which results in toxins and bacteria leaking through the mucous layer and intestinal wall to the bloodstream, is associated with GI dysfunctions and the pathology of several GI inflammatory conditions, including IBS [156] and IBD [157]. Preliminary results indicate that intestinal barrier permeability is altered in PD patients [19,20,156,158]; however, the extent and precise location of this altered barrier integrity (small intestine vs. colon) remains controversial. Nevertheless, a correlation between gut permeability and the presence of alpha-synuclein aggregates was proposed in a mouse model of PD [159], and verified in human PD samples [19], suggesting that therapeutics that target the intestinal barrier could potentially stop the prion-like spread of disease.

### 6.1. Teduglutide

Glucagon-like peptide-2 (GLP-2) is a pleiotropic hormone that stimulates mucosal growth and repair, and regulates gastric motility, gastric acid secretion, and intestinal hexose transport. GLP-2 significantly enhances the surface area of the mucosal epithelium, reduces intestinal leakiness, and delays gastric transit, thereby increasing intestinal capacity for nutrient (and potentially drug) absorption [160]. Teduglutide (Revestive) is a GLP-2 analogue that binds to GLP-2 receptors located on intestinal enteroendocrine cells and enteric neurons. Teduglutide is currently approved for use in the USA, Europe, and Australia for the treatment of short bowel syndrome. Treatment with teduglutide improves monosaccharide absorption and intestinal fluid balance. Previous studies have shown that GLP-2 receptor activation by teduglutide restores intestinal structure and functional integrity by promoting growth of the mucosa, and protects neuronal nitric oxide synthase-positive neurons, a subset of neurons important for normal gut motility [161]. Moreover, GLP-2 administration enhances the survival of enteric neurons in culture and counteracts mast cell-induced neuronal cell death (Figure 2, panel C). Thus, manipulation of GLP-2 offers a unique dual efficacy, acting as both a mucosal regenerative agent to promote nutrient absorption and as a neuroprotective agent.

Several pre-clinical studies have highlighted the efficacy of GLP-2 in mouse and cell culture models of PD [162,163]. In 1-methyl-4-phenyl-1,2,3,6-tetrahydropyridine (MPTP) lesioned mice, GLP-2 analogue administration improved the bradykinesia and movement imbalance of mice, protected dopaminergic neurons and restored tyrosine hydroxylase expression in the substantia nigra and reduced markers of inflammation and mitochondrial dysfunction [162]. Similarly, in Neuro-2a cells treated with 1-Methyl-4-phenyl-pyridine ion, GLP-2 agonist administration protected cells against mitochondrial damage, autophagy impairments and apoptosis, whilst enhancing cell signalling for mitogenesis, and reducing oxidative stress levels [163]. Whilst the impact of GLP-2 administration on GI health and function is yet to be assessed in PD, these promising findings strongly suggest a beneficial role for GLP-2 in PD. 

### 6.2. Metformin 

Anti-diabetic drugs, such as glitazones and exenatide, are known to have pleiotropic actions and have been shown to reduce the risk of developing PD in diabetic patients and reduce neurodegeneration in animal models of disease [164,165,166]. Of note, the GLP-1 receptor agonist exenatide was also found to reduce cognitive decline associated with PD [167]. Further studies are currently being undertaken to explore the therapeutic potential of exenatide in PD and other neurodegenerative conditions.

Unlike exenatide, metformin can be orally administered, which makes it a more preferential treatment than other anti-diabetic interventions. Metformin is a biguanide hyperglyceamic agent that has recently been shown to cross the blood–brain barrier [168] and protect against stroke injury and cognitive decline in diabetic patients [169,170,171]. In addition, a large population-based study showed that people on long-term metformin therapy had a reduced incidence of developing PD [172]. Although the mechanism of metformin is not clear, the drug can reduce alpha-synuclein phosphorylation, aggregation, inflammation and improve autophagy [173]. 

Given the apparent neuroprotective effect of metformin in the CNS, it is tempting to assume that metformin may protect GI dysfunction-induced enteric neuropathy. However, metformin may also protect the gut by protecting barrier integrity, with a recent study indicating that metformin modulates the gut microbiome and promotes mucus production in ageing high-fat diet mice where it was found to increase MUC2 production and increase TJ protein expression [174]. An increase in the expression levels of the TJ proteins occludin and zonula occludins-1 expression was also observed in a mouse model of colitis following treatment with metformin [175], further confirming its protective action on the gut barrier (Figure 2, panel D). 

At high doses, metformin can induce GI side-effects, such as diarrhoea and abdominal pain, which may limit its clinical application. Diarrhoea is the most common GI disturbance and is most likely caused by an increase in colonic bile salt concentrations [176], which results in abnormally high levels of water and salts entering the colon from the bloodstream. These GI side-effects are often manageable or only persist for the first few weeks after dosing; however, in 5% of cases, withdrawal is necessary [177,178]. Nevertheless, metformin’s dual action and extensive safety profile make it a promising therapeutic for improving gut health and function at the tissue and cellular level.

## 7. Targeting the Gut Microbiota 

Emerging evidence suggests that dysbiosis in the gut and subsequent alterations in the gut–brain axis may play a role in the development of motor and non-motor symptoms [179,180], including GI dysfunction. Various studies have demonstrated alterations in both α and β diversity of faecal and mucosal microbial samples from PD patients relative to controls. Whilst results are relatively heterogeneous across studies, several trends have been identified at the family and genus level, with family Verrucomicrobiaceae and genera Akkermansia and Lactobacillus increased in PD patients [181,182,183,184], while the families Prevotellaceae, Lachnospiraceae, and Pasteurellaceae and genera Blautia, Roseburia, Prevotella, and Faecalibacterium appear to be consistently decreased [181,184,185]. Further to this, a recent meta-analysis that re-analysed ten currently available 16S microbiome datasets to investigate whether there were common themes in the alterations to the gut microbiota of PD patients uncovered significant alterations in the PD-associated microbiome [21]. Enrichment of the genera Lactobacillus, Akkermansia, and Bifidobacterium and depletion of bacteria belonging to the Lachnospiraceae family and the Faecalibacterium genus, both important short-chain fatty acids producers, emerged as the most consistent PD gut microbiome alterations [21]. However, it should be noted that the microbial dysbiosis observed in PD could be a consequence of PD-related gut dysfunction (e.g., slowed gastric emptying, delayed colonic transit time) and that alleviation of these GI symptoms may subsequently result in normalization of gut flora. Whilst it is universally acknowledged that data are influenced by PD duration and severity, and several confounders need to be taken into account, these findings have seen the gut microbiota emerge as a therapeutic target in recent years. 

### Prebiotics, Probiotics, and Fecal Transplants

Several key studies have investigated the impact of prebiotic and probiotic treatment on PD-related constipation [186,187,188]. A recent double-blind, randomized, placebo-controlled, single-centre trial investigating the effects of daily multi-strain probiotics treatment in PD patients with confirmed Rome IV criteria for functional constipation found a significant difference in primary outcomes of probiotic-treated patients compared to placebo [188]. The trial found that PD patients receiving one multi-strain probiotic capsule containing eight different commercially available bacterial strains (*Lactobacillus acidophilus*, *Lactobacillus gasseri*, *Lactobacillus reuteri*, *Lactobacillus rhamnosus*, *Bifidobacterium bifidum*, *Bifidobacterium longum*, *Enterococcus faecalis*, *Enterococcus faecium*) daily for 4 weeks had significantly increased spontaneous bowel movements (1.0 ± 1.2 per week) after treatment with probiotics, compared with decreased movements in the placebo group (0.3 ± 1.0 per week, mean difference 1.3, 95% confidence interval 0.8–1.8, *p* < 0.001) [188]. Significant improvements were also seen in the secondary outcomes of changes in stool consistency, constipation severity score, and quality of life related to constipation after treatment with probiotics [188]. 

Earlier randomized, double-blind, placebo-controlled trials in PD patients with Rome III-confirmed constipation investigated the effect of fermented milk containing multiple probiotic strains and prebiotic fibre—compared with a placebo, once daily for 4–5 weeks—and uncovered that probiotic treatment resulted in a higher number of complete bowel movements (mean difference 1.1, 95% confidence interval 0.4–1.8, *p* < 0.002) [186], a significant increase in the number of days per week in which stools were of normal consistency, and significant reductions in the number of days per week in which patients felt bloated, experienced abdominal pain, or the sensation of incomplete emptying [187] (Figure 2, panels E and F).

With regard to intestinal transit time (ITT), a meta-analysis of 11 randomized controlled clinical trials with 13 treatment effects representing 464 subjects determined that short-term probiotic supplementation decreases ITT with consistently greater treatment effects identified in constipated or older adults [189]; however, the effect of probiotics on ITT has not been investigated in PD.

Fecal microbiota transplants (FMT) are also emerging a therapeutic avenue to rectify microbial dysbiosis in PD patients [190,191]. An early case report, describing the effect of FMT on a 71-year-old male patient with 7 years of resting tremor, bradykinesia and intractable constipation concluded that microbial reconstitution may have therapeutic effects in PD after the patient successfully defecated within 5 min of FMT and maintained daily unobstructed defecation until the end of follow-up (3 months). A recent evaluation of FMT in 11 PD patients with constipation found similar results with a significant reduction in the Wexner constipation score following 6 and 12 weeks of FMT therapy [191]. 

Though some doubts about the use of probiotics, prebiotics, and fecal microbiota reconstitution have been raised [192], the latest Movement Disorder Society evidence-based medicine review update on treatments for the non-motor symptoms of PD determine probiotics and prebiotic fibre as “efficacious” and “clinically useful”. However, it was highlighted that they could pose an acceptable risk without specialized monitoring [105].

The normal functioning of the gut heavily relies on a healthy gut microbiome, and an intact mucosal and epithelial barrier that inhibit potentially pathogenic toxins/microbes and their metabolites from entering the bloodstream from the lumen of the gut. When microbial dysbiosis occurs and the mucosal barrier is disrupted, pathogenic microbes can enter the blood stream through changes in tight junction protein expression that make the barrier more permeable. These pathogens can cause damage to enteric neurons located in the submucosal and myenteric plexuses, and subsequent GI dysfunction and systemic inflammation. (A) Prucalopride, (B) quercetin, (C) teduglutide, and (D) metformin all have neuroprotective properties that may protect enteric neurons from damage. In addition, these therapeutics have pleiotropic properties that involve enhancing intestinal mucous production or increasing tight junction protein expression. (E and F) Prebiotics, probiotics, and fecal transplants all aim to improve gut health and function by promoting a healthy gut microbiome. Modified from [193] and adapted from “Jejunum Epithelium”, by BioRender.com (accessed on 17 January 2022) (2021). Retrieved from https://app.biorender.com/biorender-templates (accessed on 17 January 2022).

## 8. Conclusions

The incidence of PD is rising exponentially as life expectancy increases, and the number of individuals suffering from PD is conservatively estimated to increase to 12 million worldwide by 2050 [194]. Despite PD being recognized as a multi-system disorder, and growing awareness of the tremendous impact of non-motor symptoms on quality of life and disease management, the development of therapeutics to treat these debilitating symptoms has been largely overlooked. 

Although not all PD patients have identical pathogenic mechanisms underlying their disease, GI dysfunctions are a common occurrence that require further, and better investigation. Many PD-related GI symptoms manifest early on in PD pathogenesis, and accumulating research shows that the GI tract may contribute to the spread of disease. GI function is highly dependent on gut health, which is tightly linked to intestinal gut barrier integrity, a healthy gut microbiome, and the proper functioning of enteric neurons responsible for the contraction and relaxation of the GI tract, all of which are compromised to some degree in PD patients. Therefore, using drugs that can target one, or all, parameters may prove to be beneficial and provide long-term relief for the GI symptoms that plague PD patients.

In this review, we have highlighted several potential therapeutics that either directly or indirectly target the GI tract at the tissue or cellular level. Although limited data is available on their effectiveness in PD-related GI dysfunctions, their efficacy in the CNS or other GI disorders warrants further investigation. In addition, the potential of repurposing drugs that have shown promise in the CNS or have pleiotropic properties may provide a faster identification of, and more cost-effective, therapeutics. Given the GI tract is integral for drug pharmacodynamics, improving gut health and potential drug metabolism is not only relevant for PD patients, but also other diseases/disorders where the GI system is affected and where oral administration of medications are used to treat disease. Thus, improving gut health has broader applications, and identifying drugs that can target one, or all parameters of gut heath and function will be highly beneficial to society overall.

## Figures and Tables

**Figure 1 jpm-12-00144-f001:**
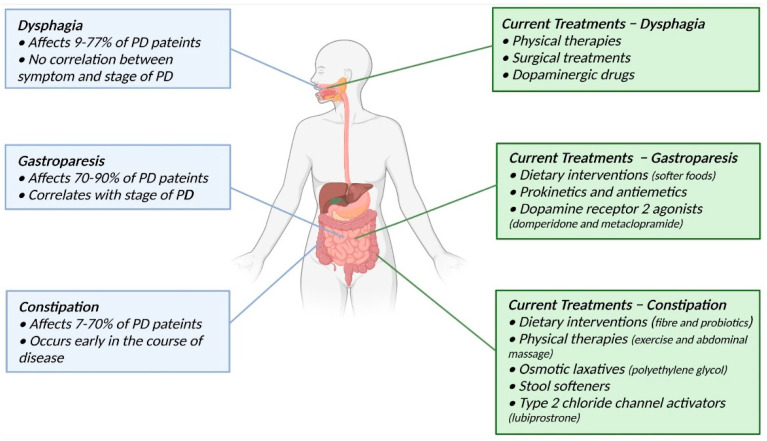
Common PD-related GI dysfunctions and current treatments.

**Figure 2 jpm-12-00144-f002:**
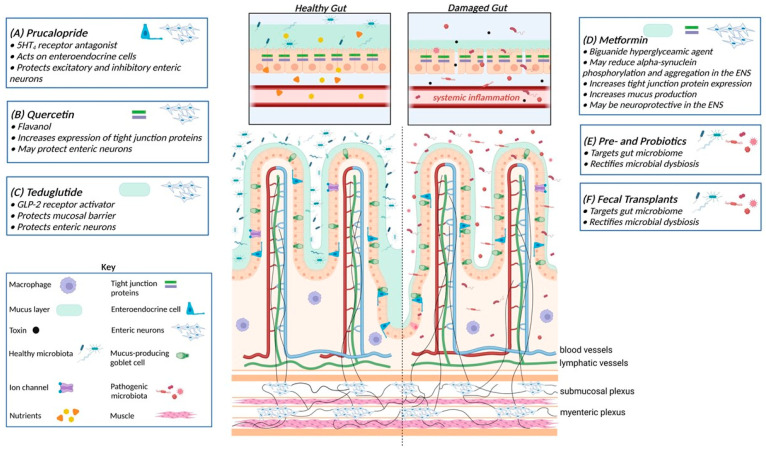
Potential new therapeutics for PD-related GI dysfunctions and sites of action.

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
