# Peer review of "Gastrointestinal Dysfunction in Parkinson’s Disease: Current and Potential Therapeutics"

_jpm, 2022, doi:10.3390/jpm12020144_

Round 1
Reviewer 1 Report
In this review, the authors discuss the main GI dysfunctions in PD patients and the novel therapies to treat GI symptoms in Parkinson's disease patients. I do appreciate this interesting and well-written review. I just recommend the following corrections:
- Abbreviations: please use only one way to report abbreviations (i.e. if you decide to put abbreviations in the brackets, do this for all of them)
-Introduction: Line 1 use Alzheimer's disease instead of dementia"
Author Response
Thank you for your comments. Please find below our point-by-point responses.
Reviewer 1:
In this review, the authors discuss the main GI dysfunctions in PD patients and the novel therapies to treat GI symptoms in Parkinson's disease patients. I do appreciate this interesting and well-written review. I just recommend the following corrections:
1) Abbreviations: please use only one way to report abbreviations (i.e. if you decide to put abbreviations in the brackets, do this for all of them)
Response: This has been corrected.
2) Line 1 use Alzheimer's disease instead of dementia"
Response: ‘dementia’ has been replaced with ‘Alzheimer’s disease’.
Reviewer 2 Report
This article presents a new valuable approach to review recent advancements in gastrointestinal dysfunction. In my opinion, the organizing, structure and writing style of the manuscript are appropriate. Coherency and integrity of the materials are also satisfying and makes the text easy-to- follow up. In addition, a great deal of recent publications related to the subject has been overviewed. I believe that this review can capture the attention of a broad spectrum of audiences and is worthy to be published after some revisions as follows:
- The quality of all the figures and graphics is not acceptable. In many cases, the writings inside the figures are not fairly readable! Try to elevate all the figures and schemes throughout the manuscript. Draw the schemes yourself for obtaining better quality and resolution!
- Highlight the differences between this review article (including materials, classification, approaches and organizing) and the other comparable review articles at the end of the introduction section.
- Be careful that abstract and conclusions are the manuscript vitrines! Try to make them more attractive for the readers! Highlight the prominent findings of this survey (e.g., most important achievements in different sub-fields) in both abstract and conclusions in brief.
- Explain in conclusions and perspectives that how can this review be useful for the broad spectrum of researchers working in different biomedical engineering fields.
- Recheck and revise the references formatting (inside the text and at the end of the manuscript) to meet the required standards.
Author Response
Thank you for your comments. Please find below our point-by-point responses, which we hope you will find satisfactory.
Reviewer 2:
This article presents a new valuable approach to review recent advancements in gastrointestinal dysfunction. In my opinion, the organizing, structure and writing style of the manuscript are appropriate. Coherency and integrity of the materials are also satisfying and makes the text easy-to- follow up. In addition, a great deal of recent publications related to the subject has been overviewed. I believe that this review can capture the attention of a broad spectrum of audiences and is worthy to be published after some revisions as follows:
1) The quality of all the figures and graphics is not acceptable. In many cases, the writings inside the figures are not fairly readable! Try to elevate all the figures and schemes throughout the manuscript. Draw the schemes yourself for obtaining better quality and resolution!
Response: We have redesigned the figures using the software program BioRender. The font size has been made larger in both Figures 1 and 2 and the layout of the figures have been rearranged to make them more aesthetically pleasing. All figures are 300dpi JPEG files, as requested in the journal requirements.
2) Highlight the differences between this review article (including materials, classification, approaches and organizing) and the other comparable review articles at the end of the introduction section.
Response: We have reworded the final paragraph of the introduction to highlight the significance of our review: “This review describes the most common GI symptoms that affect individuals suffering from PD, and the impact they have on quality of life and disease management. Given the limitations of current treatments, we propose a novel approach to the alleviation of GI symptoms by highlighting the intestinal barrier, ENS, and gut microbiome as potential therapeutic avenues, and expand on current emerging therapies and propose several therapeutics that may prove effective through drug repurposing.”.
3) Be careful that abstract and conclusions are the manuscript vitrines! Try to make them more attractive for the readers! Highlight the prominent findings of this survey (e.g., most important achievements in different sub-fields) in both abstract and conclusions in brief.
Response: We have expanded the abstract to better describe the contents of our review and believe it is now more attractive to the reader: “Unlike the central nervous system (CNS), where PD pathology and the mechanisms involved in CNS damage are relatively well characterised, the effect of PD at the cellular and tissue level in the enteric nervous system (ENS) remains unclear, making it difficult to alleviate or reverse GI symptoms. However, the resurgence of interest in understanding how the GI tract is involved in various disease states, such as PD, has resulted in the identification of novel therapeutic avenues. This review focuses on common PD-related GI symptoms and summarises the current treatments available and their limitations. We propose that targeting the intestinal barrier, ENS, and/or the gut microbiome may prove successful in alleviating PD-related GI symptoms, and discuss emerging therapies and potential drugs that could be repurposed to target these areas.”.
We have also altered the conclusion to better summarise the significance of our review: “In this review we have highlighted several potential therapeutics that either directly or indirectly target the GI tract at the tissue or cellular level. Although limited data is available on their effectiveness in PD-related GI dysfunctions, their efficacy in the CNS or other GI disorders warrants further investigation. In addition, the potential of repurposing drugs that have shown promise in the CNS or have pleiotropic properties may provide faster identification of, and more cost-effective, therapeutics. Given the GI tract is integral for drug pharmacodynamics, improving gut health and potential drug metabolism is not only relevant for PD patients, but also other diseases/disorders where the GI system is affected and where oral administration of medications are used to treat disease. Thus, improving gut health has broader applications and identifying drugs that can target one, or all parameters of gut heath and function will be highly beneficial to society overall.”.
4) Explain in conclusions and perspectives that how can this review be useful for the broad spectrum of researchers working in different biomedical engineering fields.
Response: We have expanded the conclusion to highlight how improving gut health overall has broader applications. Please see above response to comment 3.
5) Recheck and revise the references formatting (inside the text and at the end of the manuscript) to meet the required standards.
Response: All references have been rechecked, and the references have been formatted according to the MDPI ACS Journal endnote reference style file provided on the MDPI journal website.